# A Closer Look at Learned Optimization: Stability, Robustness, and Inductive Biases

**James Harrison, Luke Metz, Jascha Sohl-Dickstein**
Google Research, Brain Team
{jamesharrison, lmetz, jaschasd}@google.com

## Abstract

Learned optimizers—neural networks that are trained to act as optimizers—have the potential to dramatically accelerate training of machine learning models. However, even when meta-trained across thousands of tasks at huge computational expense, blackbox learned optimizers often struggle with stability and generalization when applied to tasks unlike those in their meta-training set. In this paper, we use tools from dynamical systems to investigate the inductive biases and stability properties of optimization algorithms, and apply the resulting insights to designing inductive biases for blackbox optimizers. Our investigation begins with a noisy quadratic model, where we characterize conditions in which optimization is stable, in terms of eigenvalues of the training dynamics. We then introduce simple modifications to a learned optimizer's architecture and meta-training procedure which lead to improved stability, and improve the optimizer's inductive bias. We apply the resulting learned optimizer to a variety of neural network training tasks, where it outperforms the current state of the art learned optimizer—at matched optimizer computational overhead—with regard to optimization performance and meta-training speed, and is capable of generalization to tasks far different from those it was meta-trained on.

## 1 Introduction

Algorithms for stochastic non-convex optimization are a foundational element of modern neural network training [1]. Choice of optimization algorithm (and the associated hyperparameters) is critical for achieving good performance of the underlying model, as well as training stability. Practically, there are few formal rules for choosing optimizers and hyperparameters, with researchers and practitioners typically defaulting to a small number of optimizers and associated hyperparameters with which they are familiar. Moreover, the algorithms chosen between are usually derived based on analysis in the convex setting or informal heuristics, and algorithm performance varies substantially across different real world tasks [2].

To resolve these issues, *learned optimizers* have been proposed [3–8]. These optimizers are typically either composed of blackbox function approximators (such as neural networks) or hand-designed functions containing hyperparameters that are learned. The distinguishing factor of this class of optimizers is the training methodology: because global search over hyperparameters is computationally intractable for optimizers with many parameters, local gradient-based methods are used [7–9]. Thus, the *meta-training* of these optimizers—in which they are trained to optimize some metric of performance for the underlying model such as validation loss—has strong similarities to training other neural network models.

While learned optimizers are potentially transformative, they suffer from several fundamental flaws preventing their broad uptake. The first is reduced performance and stability when they are applied in circumstances unlike those in which they are meta-trained—for instance, learned optimizers often

36th Conference on Neural Information Processing Systems (NeurIPS 2022).

diverge when used to train models for more steps than the learned optimizer was applied for during meta-training [10–12]. The second is instability and inconsistency in the meta-training of the learned optimizers themselves—learned optimizer performance is often highly dependent on random seed, and meta-training trajectories can get stuck for many steps and make inconsistent progress [8, 9].

In this paper, we:

- Use tools from dynamical systems theory to characterize the stability of the parameter dynamics induced by learned optimizers, via eigenvalue analysis of training in the noisy quadratic setting.

- Propose a series of changes to the architecture and training of learned optimizers that improve their stability and inductive bias. These changes include incorporating a tunable contribution from a nominal optimizer with descent guarantees, using large magnitude weight decay on the optimizer's parameters, and preconditioning the updates generated by the learned optimizer before applying them to the parameters.

- Demonstrate experimentally that the resulting *stabilized through ample regularization* (STAR) learned optimizer is more stable and faster to meta-train, is more stable and performant than baseline learned optimizers even when applied for many more training steps than used during meta-training, and generalizes well to new tasks which are dissimilar from tasks it was meta-trained on. For instance, STAR generalizes well to a transformer task with $175\times$ more parameters and $5\times$ the number of training steps than the MLP it was meta-trained to optimize. Effective learned optimizer generalization after meta-training on a single task has not previously been demonstrated.

## 2   Related Work

Learned optimization has seen a recent surge of interest motivated by the success of deep learning methods in a wide variety of problems [3–8, 13–16]. These methods fit into a larger class of *meta-learning* methods, which aim to leverage the success of expressive learning algorithms (such as neural networks) to learn *learning algorithms* [17–19]. Meta-learning algorithms have been particularly successful and widely investigated in the domain of few-shot learning, in which an agent must learn to make predictions based on a small amount of data. Early approaches to this problem focused on blackbox models such as recurrent networks [20–23] due to their expressive power. However, works that integrate algorithmic inductive biases—for example by exploiting gradient descent [24, 25], metric learning [26, 27], convex optimization [28, 29], exact or approximate Bayesian inference [30–32], changepoint detection [33, 34], and other algorithmic primitives—have been highly successful when applied to few-shot learning. However, similar investigation of inductive biases for meta-learning beyond the few-shot learning setting has been largely absent.

The difficulty of stable training of large models led to the development of adaptive optimization algorithms (such as Adagrad [35], RMSProp [36], Adam [37], and many other) which are relatively invariant to hyperparameter choice. In practice, however, tuning of these hyperparameters is still necessary both over the course of training and across optimization problems. While further methods have been developed for automatic online hyperparameter tuning [38–40], a line of work has focused on meta-learning a neural network that chooses hyperparameters as a function of training history [41–45]. This approach results in inherently better stability of learning algorithms as the output is typically restricted to (in expectation) descent directions. In this work, we find such inherent stability is crucial for guiding training, especially early in meta-training.

Shortly after the advent of deep learning, blackbox optimizers were developed which aimed to exploit the expressivity of neural networks in optimizer design. Of particular relevance to this work are [46, 47], which use reinforcement learning (combined with guided policy search [48]) and the approaches taken in [6, 7] which are directly trained via backpropagation through time [49]. The guided policy search learning strategy is one of several methods that leverage forms of curriculum learning to stabilize learning [50]. These works all propose networks that ingest (a history of) gradients and return a parameter updates. As investigated by [8], the long computational graphs necessitated by meta-training can result in chaotic behavior which makes meta-training extremely noisy. Numerous approaches to address this problem have been proposed. [8, 51] propose truncated zeroth order optimization which avoids extremely noisy reparameterization gradients combined with truncated computation graphs. [45–47] among others use reinforcement learning, in which the

truncated computational graph is augmented with a value function capturing dependency of future losses on the policy. [7, 9] leverage input features computed via fixed momentum operators which yield stable-by-design hidden states. In this paper we dive into the question of stability of meta-training, and find that carefully designing stability into blackbox optimizers is both necessary, and yields optimizers that both outperform the prior state of the art (at matched optimizer computational overhead) and improve overall stability.

# 3   Problem Statement

We will consider the problem of training a neural network by optimizing its parameters $\phi \in \Phi \subseteq \mathbb{R}^N$. We aim to minimize the expectation over data of loss function $\ell : \Phi \times \mathcal{X} \to \mathbb{R}$ which acts on parameters $\phi$ and data (point or minibatch) $x \in \mathcal{X}$. We define our expected loss at iteration $t$ as

$$\mathcal{L}_t(\phi) = \mathbb{E}_{x_t}[\ell(x_t; \phi)]. \tag{1}$$

A learned optimizer is defined by a parameteric update function $f(\cdot; \theta)$ with meta-parameters $\theta \in \Theta$, which acts on a history of training statistics such as parameters, loss values, gradients, and training iteration number. We write these combined input features at time $t$ as $z_t$. When used to train a model with parameters $\phi$, then $f(\cdot; \theta)$ maps input state $z_t$ (as well as optimizer hidden state which we do not explicitly write) to updated parameters $\phi_{t+1}$, with an update of the form

$$\phi_{t+1} = \phi_t - f(z_t; \theta). \tag{2}$$

In this paper we consider only the problem of optimizing the train loss of the inner problem. A particular strength of learned optimizers is that they can be trained with respect to validation loss [8, 45]. However, this capability is independent and complementary to the topics of this paper, so following Metz et al. [9] we focus on train loss to remove factors of variation from experimentation. Our goal in meta-learning is thus to find the parameters $\hat{\theta}$ that minimize the meta-loss $L(\theta; T)$,

$$\hat{\theta} = \arg\min_{\theta \in \Theta} L(\theta; T), \qquad L(\theta; T) = \sum_{t=1}^{T} w_t \mathcal{L}_t(\phi_t), \tag{3}$$

under the inner parameter dynamics imposed by (2), where $w_t$ is a weighting term (as has been used previously, e.g. [6]), and where $T$ denotes a maximum (possibly infinite) run length.

# 4   Understanding Optimizer Performance: The Noisy Quadratic Setting

In this section, we examine performances differences between optimizers, with a particular focus on the impacts on meta-learning of learned optimizers. We characterize the behavior of optimizers in the *noisy quadratic* setting. This setting—consisting of a quadratic optimization problem with randomly sampled, i.i.d. minima—aims to be representative of optimization with stochastic minibatches [11, 52, 53]. Moreover, this setting is a reasonable proxy for wide neural networks [54, 55].

The loss[1] at each timestep is

$$\ell(\phi_t) = \frac{1}{2}(\phi_t - \xi_t)^\top H (\phi_t - \xi_t) \tag{4}$$

with i.i.d. $\xi_t \sim \mathcal{N}(0, \Sigma_\xi)$, yielding gradient $\nabla_t = \nabla_{\phi_t} \ell(\phi_t) = H(\phi_t - \xi_t)$. Our chosen loss is the average of the loss at each timestep, and thus $w_t = 1/T$ for all $t$.

We consider an update of the form

$$\phi_{t+1} = \phi_t - (g_t + P\nabla_t) \tag{5}$$

where $g_t$ corresponds to a *nominal* term, which we do not aim to meta-learn, and $P$ corresponds to our meta-learned term (thus corresponding to $\theta$ in the notation of the previous section), here taking the form of a dense preconditioner matrix. In this section, we will primarily consider nominal terms of the form $g_t = \alpha \nabla_t$, yielding combined autonomous dynamics

$$\phi_{t+1} = \underbrace{(I - (\alpha I + P)H)}_{A} \phi_t + \underbrace{(\alpha I + P)H}_{I-A} \xi_t. \tag{6}$$

---

[1] Note that it is also common to consider a deterministic loss of the form $\phi_t^T H \phi_t$ with a noisy gradient $\nabla_t = H\phi_t + \xi_t$. These settings are roughly equivalent, with our setting accumulating an extra $\mathrm{tr}(H\Sigma_\xi)$ at each timestep in expectation.

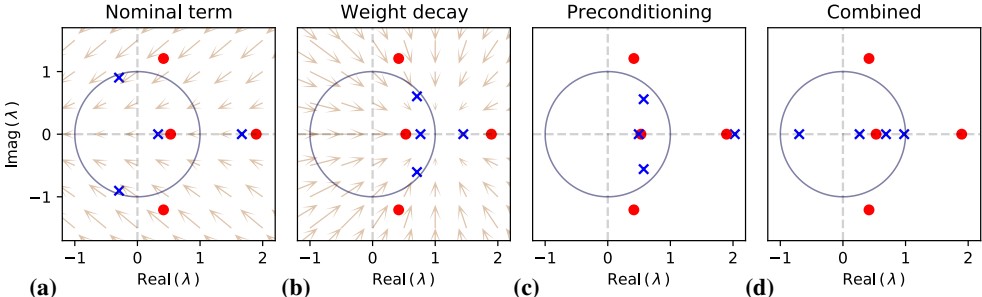

Figure 1: **Nominal terms, preconditioning, and weight decay improve the inductive bias of learned optimizers.** We visualize eigenvalues of the induced parameter update map $A$ (Equation 6) when a simple linear learned optimizer is applied to train a quadratic model. Eigenvalues within the unit circle correspond to stable optimization, while those outside the unit circle will lead to diverging training trajectories. Red circles and blue Xes correspond to eigenvalues before and after making each intervention, respectively. Arrows show how an intervention will tend to shift eigenvalues as the intervention magnitude is increased. **(a)** Incorporating a nominal term causes eigenvalues to have a more negative real part, and to decay toward the real line. This can cause training dynamics to remain stable even when the learned optimizer alone would cause gradient ascent in some direction. **(b)** Increasing weight decay pulls the entries in $P$ toward 0, turning off the learned optimizer and thus pulls the eigenvalues of $A$ toward 1. **(c)** Pre-conditioning reduces the impact of the problem's Hessian $H$. This leads to eigenvalues that depend more on properties of the optimizer, and less on properties of the problem being optimized. **(d)** Through a weighted combination of the three interventions of (a-c) all eigenvalues are mapped into the unit circle, and the learned optimizer becomes stable.

## 4.1 Nominal Terms Shift the Region of Stability

Central to our discussion will be the notion of stability. In the study of deterministic dynamical systems (for example, taking $\Sigma_\xi \to 0$), asymptotic stability[2] is guaranteed in linear system (6) when $\rho(A) = \max_i |\lambda_i(A)| < 1$ for all eigenvalues $\lambda_i(A)$ of $A$. Moreover, for the linear dynamical system we consider in this section, asymptotic stability implies $\lim_{T\to\infty} L(\theta; T)$ is finite, whereas instability implies $\lim_{T\to\infty} L(\theta; T) = \infty$ for both the deterministic and stochastic system. Thus, stability is a necessary condition for achieving optimizers which, in expectation, reduce the loss over long training horizons. However, we emphasize that stability here is a proxy for convergence that yields simplified discussion and analysis in our setting.

While stability is essential for guaranteeing favorable optimizer performance, the noisy quadratic setting also gives us insight into the relationship between the stability of underlying dynamical system and the gradient with respect to meta-trained optimizer $P$. This gradient is essential for meta-training the optimizer. Under the dynamics (6), the parameters at time $t > 0$ are $\phi_t = A^t\phi_0 + \sum_{k=0}^{t-1} A^{t-k-1}(I - A)\xi_k$ and thus $\phi_t \sim \mathcal{N}(A^t\phi_0, \Sigma_t)$ where

$$\Sigma_t = \sum_{k=0}^{t-1} A^{t-k-1}(I - A)\Sigma_\xi (A^{t-k-1}(I - A))^\top. \tag{7}$$

Thus, our expected loss is

$$L(\theta; T) = \frac{1}{T}\sum_{t=1}^{T}(\phi_0^\top (A^t)^\top H A^t \phi_0 + \operatorname{tr}(H(\Sigma_t + \Sigma_\xi))). \tag{8}$$

This loss at time $t$ is polynomial in $A$ with degree $2t$ (and note that $A$ is constant across time). Thus, the gradient of this loss term with respect to $P$ is polynomial in the parameters of $A$ with degree $2t - 1$. As a result, instability of the dynamics of $\phi$ generally implies instability of the gradient of the loss with respect to $P$, which in turn implies the expected gradient magnitude and gradient variance both diverge for long horizons.

---

[2]Asymptotic stability is defined by limiting behavior converging to a constant, in our case chosen to be 0, $\lim_{t\to\infty} \phi_t = 0$. Since our noisy quadratic model has a global minimum at 0, this corresponds to reaching minimum loss. See Appendix A.1 for more discussion of stability and related minor technical results.

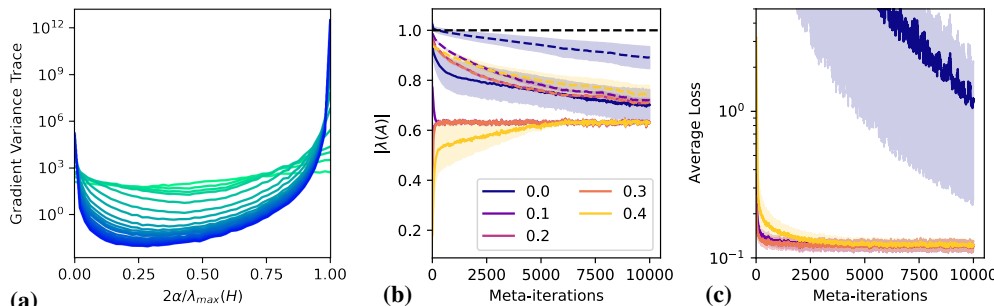

Figure 2: **Nominal terms improve the stability and trainability of learned optimizers. (a)** We adjust the coefficient of the nominal term for the optimizer described in Equation 7 from 0 to $2/\lambda_{\max}(H)$. The plot shows the trace of the variance of the meta-parameter gradient as a function of $\alpha$, with different colors correspond to trajectory lengths from $T = 1$ (green) to $T = 1000$ (blue). For $\alpha$ within our stability bounds, we achieve lower variance gradient estimates for all problem lengths $T$, with particularly dramatic reduction for larger $T$. **(b)** The magnitude of the eigenvalues of the dynamics matrix, $A$, as a function of meta-iteration during meta-training. These are shown for $2\alpha/\lambda_{\max}(H) \in \{0, 0.1, 0.2, 0.3, 0.4\}$, corresponding to the legend items. Lower variance gradients in pane (a) correspond to more stable, smaller magnitude, eigenvalues. **(c)** The meta-loss (here corresponding to the mean loss up to $T = 50$) for varying $\alpha$. Colors correspond to the legend of the center figure. Improving the inductive bias of the learned optimizer, by adjusting $\alpha$, leads to near-optimal behavior in under 1000 meta-training iterations, while poorly chosen biases may require $100\times$ as many iterations, or fail to train entirely. See Appendix B.1 for experimental details.

How can we guarantee stability of the update dynamics? We have stated previously that we require[3] $\rho(A) < 1$, which we can map to eigenvalues of $P$ via the following result. We make the technical assumption that $P$ has real eigenvalues and is diagonalizable; this is satisfied in the case that $P$ is symmetric as is common for preconditioner matrices. This assumption allows us to strictly order eigenvalues of $A$, and we write $\lambda_{\min}(A) = \min_i \lambda_i(A)$ and $\lambda_{\max}(A) = \max_i \lambda_i(A)$. More importantly, this assumption clarifies the presentation in this primarily pedagogical discussion.

**Theorem 1.** *Let $A = I - \alpha H - PH$ with $H$ symmetric positive definite, $P$ diagonalizable with real eigenvalues, and $\alpha \geq 0$. Then $\rho(A) \leq 1$ if*

$$-\alpha \leq \lambda_{min}(P) \tag{9}$$

$$\lambda_{max}(P) \leq \frac{2}{\lambda_{max}(H)} - \alpha \tag{10}$$

The proof of this result and all other results is provided in the Appendix A.2. There are a few key takeaways from this result. The lower bound (9) highlights the stabilizing effect of the nominal optimization term. If a particular learned optimizer would result in divergence due to moving uphill in a particular direction (corresponding to the eigenvector associated with the minimum eigenvalue) without the nominal term, the addition of the nominal term $\alpha \geq 0$ gives us additional stability margin. The tradeoff is that for the upper bound (10) we lose stability margin. Thus, we can in general learn only smaller $P$ (as measured by the induced spectrum of $A$).

In the design of a learned optimizer, the addition of the nominal term can lead to especially large steps which result in divergence due to violation of (10). As such, we must be cautious to limit the magnitude of $\alpha$, and this result motivates regularization of $P$. Within the quadratic setting, we are unlikely to practically hit this upper bound as it corresponds to taking steps large enough to oscillate to divergence. For neural network optimization, however, optimization tends to predictably (albeit approximately, due to minibatch dynamics) hit this upper bound [56].

We illustrate the effect on eigenvalues of incorporating a nominal term in Figure 1. We investigate the stability properties of the noisy quadratic model in Figure 2, where we find the nominal term makes learned optimizers more stable, and easier to train.

---

[3]Note that our analysis will include the case where $\rho(A) = 1$, implying *marginal* stability in which the system neither converges nor diverges.

## 4.2 Preconditioners can Stabilize and Simplify the Design of Update Dynamics

Adaptive preconditioners have seen widespread use in large-scale machine learning. Methods such as Adagrad [35], RMSProp [36] and Adam [37] are some of the most fundamental tools in training neural networks. These approaches typically approximate the inverse square root Hessian, and apply a transformation

$$\tilde{g}_t = H^{-\frac{1}{2}} g_t \tag{11}$$

where $g_t$ is some nominal optimizer. Typically this approximation is diagonal which corresponds to maintaining some weighted recursive estimate of the magnitude of each element of $g$. This transformation roughly normalizes the step size, making steps isotropic and thus the effective learning rate may be better controlled.

If we combine this Hessian preconditioner together with our preconditioner $P$ and nominal step size $\alpha$, the resulting dynamics are

$$\phi_{t+1} = \phi_t - H^{-\frac{1}{2}}(\alpha I + P)\nabla_t. \tag{12}$$

Stability in this preconditioned setting is given by the following result.

**Lemma 2.** *Consider parameter dynamics given by (12) with assumptions on $P, H$ matching Theorem 1. Then (12) is stable if (9) holds for (12) and*

$$\lambda_{max}(P) \leq \frac{2}{\sqrt{\lambda_{max}(H)}} - \alpha. \tag{13}$$

Note, this results in a looser upper bound than (10) if $\lambda_{\max}(H) > 1$.

Why apply the preconditioner to the output of the learned optimizer as opposed to the input gradient? Normalization at the output of the learned component results in better robustness than normalizing the input, as it is (relatively) robust to arbitrary initializations of the blackbox term. We expand on this in both Section 4.3 (for quadratic problems) and Section 5 (for general learned optimizers).

## 4.3 Adaptive Nominal Terms Improve Robust Stability

We have so far motivated choosing a nominal $\alpha > 0$ to bias the optimizer dynamics toward descent/stability. However, it is likely that we do not want to leave $\alpha$ fixed over the course of inner or outer (meta) training. In inner training, decreasing the learning rate over the course of training has been shown to improve empirical performance (and is often necessary for guaranteeing convergence [57]), and is almost always done when training neural networks.

For simple nominal gradient estimators and comparably complex blackbox terms, the blackbox model should be able to cancel out the nominal term, and induce the effects of reducing $\alpha$. Setting $P = P^* - \alpha I$ with nominal gradient term $g_t = \nabla_t$ yields closed loop dynamics

$$\phi_{t+1} = \phi_t - \alpha\nabla_t - (P^* - \alpha I)\nabla_t = \phi_t - P^*\nabla_t \tag{14}$$

which is optimal with respect to meta-loss for optimal $P^*$. In this subsection, we argue from the point of view of robust stability that this strategy is suboptimal relative to direct control of the magnitude of the nominal and blackbox term.

We introduce a multiplicative error model[4] which captures the sub-optimality of the learned $P$ during meta-training. Let $P = \Delta\tilde{P}$ for diagonal disturbance $\Delta \in \mathcal{D}$. We define this uncertainty set as

$$\mathcal{D} = \{\Delta \in \mathbb{R}^{N \times N} : \Delta = \text{diag}(d), \ 0 < d_i \leq \overline{d}_i, \ d \in \mathbb{R}^N\}. \tag{15}$$

Within this error model, we can establish conditions for stability in line with the previous subsections. We wish to establish *robust stability* conditions, which guarantee the stability of the dynamical system for all realizations of the disturbance.

---

[4]Our diagonal multiplicative error model is related to a standard formulation within the analysis of the robust stability of linear dynamical systems, known as D-stability [58–60], although we emphasize that D-stability analysis is usually in continuous time.

**Lemma 3.** *Let $A = I - \alpha H - PH$ with $P = \Delta \tilde{P}$. We will assume $\tilde{P}$ and $\Delta$ are simultaneously diagonalizable, $H$ symmetric positive definite, and $0 < \alpha < 2/\lambda_{max}(H)$. Then, $\rho(A) \le 1$ for all $\Delta \in \mathcal{D}$ if*

$$-\frac{\alpha}{\max_i \overline{d}_i} \le \lambda_{min}(\tilde{P}) \tag{16}$$

$$\lambda_{max}(\tilde{P}) \le \frac{1}{\max_i \overline{d}_i} \left( \frac{2}{\lambda_{max}(H)} - \alpha \right) \tag{17}$$

These results generally result in the tightening of the margin of stability. If we choose $\tilde{P} = P^* - \alpha I$, corresponding to our previously discussed cancellation of the nominal term, our dynamics become

$$\phi_{t+1} = \phi_t - (\alpha(I - \Delta) + \Delta P^*)\nabla_t, \tag{18}$$

which harms the stability margins. This is well understood by taking $\Delta = (1 - \epsilon)I$, for any adversarial $\epsilon \le 1$. Then, we have update dynamics

$$\phi_{t+1} = \phi_t - (\alpha \epsilon - (1 - \epsilon)P^*)\nabla_t \tag{19}$$

yielding dynamics matrix $A = I - \alpha \epsilon H - (1 - \epsilon)P^* H$. Here, $\alpha \epsilon H$ is the excess term in the dynamics compared to if the nominal term had instead been set to 0. In this expression, $\epsilon$ adversarially perturbs the system toward instability, resulting in a potentially substantial performance drop.

Instead, we can select $\alpha$ over the course of both inner and outer training via directly controlling the magnitude of $\alpha, P$. This approach corresponds to a hyperparameter controller, which has both been shown to enable automatic control of step sizes [45] over the course of inner training.

### 4.4 Non-Markovian Optimizers Require Joint Stability

We briefly discuss the role of non-Markovian (or hidden state) dynamics in learned optimizers. An extended discussion is in Appendix A.4. The role of momentum and stable hidden states has been investigated in detail in both works on optimization [53, 56] and learned optimization [61]. We summarize the key points below.

The core analysis change required is that we must consider the joint stability of the hidden state dynamics and the parameter dynamics together. Such analysis has been done in the case of Polyak momentum [62], and has been found to yield less restrictive upper stability margins (thus allowing a larger step size). In the general case of blackbox optimizer dynamics, the stability properties of the hidden state update have been investigated in [61], and we expand on this in Section 5.

Momentum often accelerates convergence in the full-batch optimization setting. However, it has the additional benefit of filtering stochastic gradients, yielding better expected descent. This filtering behavior is an important consideration in any learned optimizer operating in the stochastic setting.

## 5 Designing a Better Learned Optimizer

In this section, we present a set of regularization strategies and architectural modifications for learned optimizers that leverage the insights gleaned in the previous section. We refer to our optimizer as a *stabilized through ample regularization* (STAR) learned optimizer[5]. This section also presents a limited selection of experiments on in-meta-distribution and out-of-meta-distribution performance, with experimental details available in Appendix B and more results available in Appendix C.

### 5.1 New Design Features in the STAR Optimizer

**Bias toward descent.** We add a nominal term, as described in Section 4.1. We focus on nominal terms based on Adam [37] and AggMo [63], in part due to the input features to our blackbox optimizer containing momentum at different timescale (as in AggMo) and running exponential moving average (EMA) gradient magnitude estimates (as in Adam). We experimentally compare different nominal terms in the appendix.

---

[5]The code for our optimizer is available here: `https://github.com/google/learned_optimization/blob/main/learned_optimization/learned_optimizers/adafac_nominal.py`

We add a magnitude controller on the nominal descent term, following the discussion in Section 4.3. This magnitude controller consists of one additional output head with an exponential nonlinearity on the small MLP (requiring five additional weights). The combination of magnitude control applied to a nominal optimizer (such as Adam) makes our full nominal term equivalent to a hyperparameter-controller learned optimizer, albeit with a controller that leverages substantial computation reuse with the blackbox term.

**Magnitude control via weight decay.** In order to discourage violations of the upper bound on stable eigenvalues in Section 4.1, we apply relatively heavy weight decay ($L_2$ regularization) in outer training to the parameters of the blackbox term. By directly controlling weight magnitudes (as opposed to controlling magnitudes via regularizing network outputs) we achieve magnitude regularization for arbitrary inputs (for reasonably-sized inputs) and thus achieve better generalization.

**Preconditioner-style normalization.** As discussed, we use an adaptive inverse EMA preconditioner in our nominal term to better bias our model toward descent. As suggested by Section 4.2, we apply the same preconditioner to the output of the blackbox term as well as the network inputs. Because these EMA terms are maintained as inputs to the network already, the expense of this transformation is only the cost of dividing the blackbox output by the preconditioner.

**Stable hidden states.** We discussed the importance of considering the stability of the combined parameter/hidden state dynamics in Section 4.4. It is critical to design the hidden state update dynamics to bias toward stability; indeed, this is a well-known fact in the study of recurrent networks in general [64–66] and has been discussed in [7, 61]. In this paper, as in previous works [7, 9], we use exponential moving average, momentum-style hidden states which are stable by design.

## 5.2 Overview of the STAR Optimizer

We apply these modifications to the *small_fc_lopt* optimizer presented and open sourced in [9]. This optimizer is a small (197 weight) MLP which is applied elementwise—it takes inputs such as the parameter value, the gradient, and features such as gradient momentum at different timescales, and outputs an update to the parameter. This optimizer is applied in parallel to all parameters in the model being trained, and the only interaction between parameter updates is through tensor-level input features. We refer to Appendix A of [9] for a complete explanation of the optimizer, but we review the basic details here. The optimizer is parameterized as

$$f(z_t) = \beta_1 d_\theta(z_t) \exp(\beta_2 m_\theta(z_t)). \tag{20}$$

In this expression, $\beta_1, \beta_2$ are small constants which in [9] and here are set to 0.001. The terms $d(z_t)$ and $m(z_t)$, corresponding to direction and magnitude terms respectively (so named because the magnitude term only positively scales the direction output), are heads of the neural network. The architecture of the optimizer is an MLP with two hidden layers, each with a width of four. This limits the number of total parameters to 197, yielding high computational efficiency versus most learned optimizers. The input features of the network are the parameter value, various parameter-wise momentum terms, EMA gradient norms estimates, several adafactor-based [67] features which aggregate tensor-level information, and a parameterization of the training step.

Our modified update takes the form $f(z_t) = f_b(z_t) + f_g(z_t)$ where $f_g(\cdot)$ is the nominal term and $f_b(\cdot)$ is the blackbox term. The nominal term is structured as

$$f_g(z_t) = \beta_1 \exp(\beta_2 m_g(z_t)) g(z_t) \tag{21}$$

where $m_g(\cdot)$ is a magnitude term for the nominal term and $g(z_t)$ is the nominal term; in our experiments we use a combined AggMo [63] and Adam [37] for this term. Critically, this term corresponds directly to a descent direction without being passed through a network, biasing the update toward descent. The blackbox term is structured as

$$f_b(z_t) = \beta_3 \frac{d(z_t)}{v(z_t)} \exp(\beta_4 m_b(z_t)) \tag{22}$$

where $v(z_t)$ is our specified preconditioner term and $m_b(\cdot)$ is the blackbox magnitude term. The difference between this parameterization and (20) is the normalization term. In these terms, $\beta_1, \beta_2, \beta_3, \beta_4$ are hand-specified constants, and the neural network has three output heads. The addition of the extra head as well as the multiplicative factor on that term adds an extra five parameters to the original 197-parameter optimizer, as well as one more for $\beta_1$ scaling the entire nominal term. Note that the addition of the $v(\cdot)$ term in the denominator of the blackbox term has magnitude comparable to the mean gradient magnitude, which requires choosing $\beta_3$ to account for this change in magnitude.

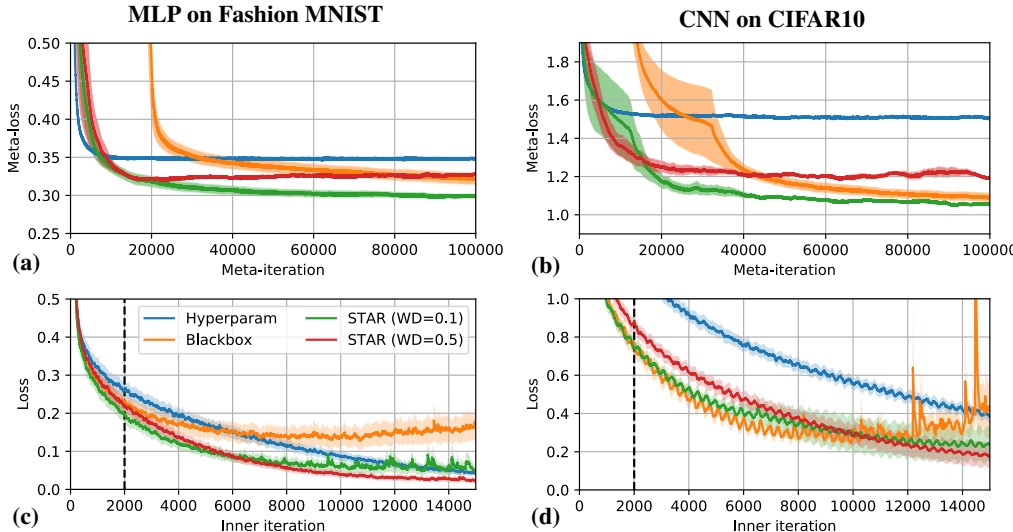

Figure 3: **The STAR learned optimizer trains faster, achieves better fully-trained optimizers, and has better stability than prior learned optimizers.** We visualize meta-training and inner training results for our STAR optimizer, at two different values of weight decay, as well as for a purely blackbox optimizer (the *small_fc_lopt* optimizer from Metz et al. [9]) and a hyperparameter controller model. The upper row shows meta-training curves for **(a)** a two hidden layer MLP on Fashion MNIST, and **(b)** a three layer CNN on CIFAR10. The lower row shows inner training curves for **(c)** Fashion MNIST, and **(d)** CIFAR10. The meta-training objective consists of the average loss over the first 2000 inner iterations, with this horizon indicated by the black dashed line. The blackbox models diverge outside of their meta-training horizon. The STAR model, on the other hand, remains stable when applied for more steps than used during meta-training.

## 5.3 The STAR Optimizer Improves Performance

As described and shown in Figure 3, the STAR optimizer shows improved stability and faster meta-training than baselines, including the *small_fc_lopt* optimizer from Metz et al. [9], which is on the performance vs. optimizer overhead Pareto frontier. We refer to *small_fc_lopt* as "Blackbox" in our experiments, and emphasize that the STAR optimizer is architecturally identical to the blackbox optimizer other than the modifications documented in the previous subsection. We also compare to a hyperparameter-controller learned optimizer ("Hyperparam"). This optimizer consists of our nominal term with the magnitude control head, but removing the blackbox term. Experimental details are provided in Appendix B and in our open source code. Experiments on additional tasks, with other values of weight decay, ablations of the primary components of the STAR optimizer, and visualization of different random seeds as opposed to aggregated statistics, are provided in Appendix C.

Most interestingly, the STAR optimizer continues to perform strongly even when optimizing for almost two orders of magnitude more steps than it was applied for during meta-training. It generalizes in this way to longer training runs without sacrificing performance on the meta-training loss. Divergence or poor performance outside of the meta-training setting (such as for the blackbox optimizer in Figure 3d) is a primary limitation to the application of learned optimizers. In contrast, the STAR optimizer has controllable and reliable stability behavior outside of the meta-train setting. While meta-generalization is helped by training across varying settings (including varying the meta-train horizon and the underlying task) [72], this *inherent* robustness—the ability of a model to generalize well to new tasks despite not being meta-trained with the explicit goal of generalization—is critical to achieving broadly applicable optimizers. We emphasize that the usual approach to meta-generalization—large-scale meta-training across many large tasks—is extremely expensive, typically requiring weeks of computation on millions of dollars of hardware. In contrast, we achieve dramatically improved stability fast and for free. We refer the reader to Metz et al. [9] for further baseline results (including heavily-tuned non-learned optimizers) on the tasks evaluated in Figure 3.

We further explore the inherent generalization abilities of the STAR optimizer in Figure 4. We apply the optimizer trained on the first task (a small MLP applied to Fashion MNIST) to a wide variety of learning tasks including large models such as a transformer [73]. Remarkably, the STAR

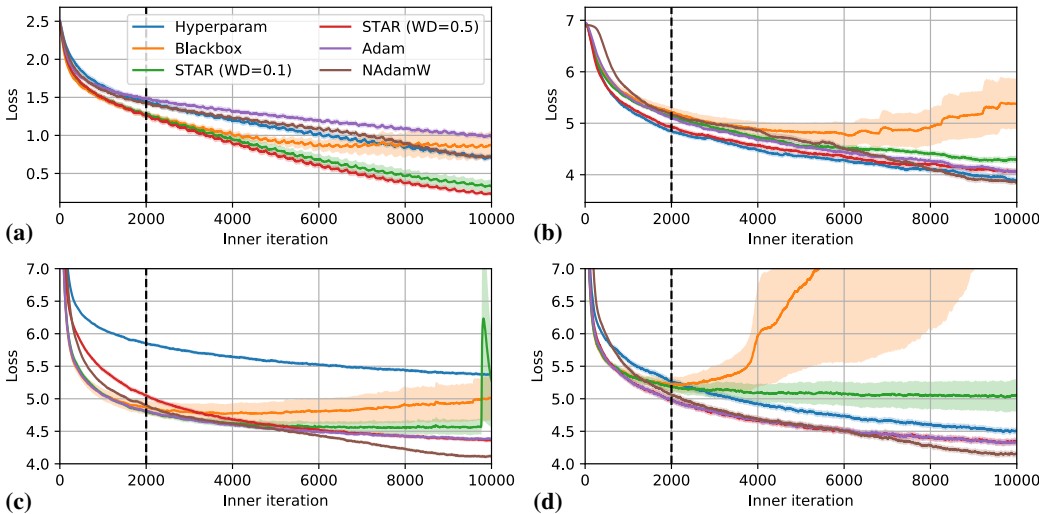

Figure 4: **After only being meta-trained to optimize an MLP on Fashion MNIST for 2000 inner-iterations, the STAR learned optimizer is able to generalize to never before seen problems.** We show performance on the following tasks: **(a)** A 3 hidden layer MLP, with layer norm, trained on Cifar10. **(b)** A shallow Resnet-like [68] model trained on 32x32 ImageNet [69]. **(c)** A 256 unit LSTM [70] language model trained on LM1B [71]. **(d)** A 5 layer, 256 hidden size decoder-only transformer (also on LM1B). In all cases, the blackbox optimizer diverges, while the STAR models, with appropriately chosen weight decay, continue to descend on the loss.

optimizer—which was trained in an extremely limited setting—generalizes well to different network sizes, nonlinearities, and datasets (including to language tasks). STAR nearly uniformly ourperforms the baseline blackbox model. The performance of STAR (for the correct choice of weight decay) is comparable to (and occasionally substantially better than, as in Figure 4a) a hyperparameter-tuned Adam model. These tasks were selected from a subset of evaluation tasks. See the Appendix for learning curves for all generalization experiments.

## 6 Discussion

This paper has addressed the role of stability and inductive biases in learned optimizers, and we have shown that incorporating stabilizing inductive biases results in strong performance both in-distribution and out-of-distribution. Much further work on injecting stability into learned optimizers remains to be done, and designing inductive biases for learned optimizers is potentially an exciting new line of work that straddles traditional optimization theory and the study of learned optimizers. There is a deep literature on the stability of general computation graphs that can, and should, be exploited to allow more fine-grained control of stability properties of optimizers [74–78].

Informally, stabilization of the computation graph (as we have defined it) is a sufficient condition to guarantee that an optimizer moves downhill on the training loss landscape (in expectation). Our motivation from this comes from the convex setting, in which always moving downhill (with appropriate technical conditions) will lead to a global minimum. However, it is not clear if our induced biases are optimal for the training of neural networks, and in general, it is unclear the extent to which neural network training resembles convex optimization problems [55, 79, 80]. Further study of which inductive biases are desirable for optimizing networks is necessary. Moreover, while we have shown strong generalization properties arise from our stability properties, there may be further desirable inductive biases that could be induced to improve generalization, such as for example biasing networks toward flat minima [81, 82].

## Acknowledgments and Disclosure of Funding

We thank Daniel Freeman and Rohan Sinha for helpful conversations and comments in the development of this work.

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
