# A    The Noisy Quadratic Setting: Additional Details

In this section we extend our discussion of the noisy quadratic model (NQM). We first discuss stability in the NQM. We then provide proofs for the results in Section 4. We also extend our discussion of robust stability and the stability of models with hidden states.

## A.1    Stability

In this subsection we expand on the short discussion of key stability results in the body of the paper. We will primarily discuss stability of the nominal system. This is motivated by a few results.

**Lemma 4.** *Consider the dynamical system defined by (6) with all eigenvalues of $H$ finite. We write the expected total loss for horizon $T$ as $L(\theta; T)$. Consider also the deterministic system given by (6), in which $\Sigma_\xi \to 0$. We will write the total loss under these dynamics as $\overline{L}(\theta; T)$. Then,*

$$\overline{L}(\theta; T) \leq L(\theta; T) \tag{A.1}$$

*for all $T > 0$.*

*Proof.* Note that

$$\overline{L}(\theta; T) = L(\theta; T) + \frac{1}{T} \sum_{t=1}^{T} \operatorname{tr}(H(\Sigma_t + \Sigma_\xi)). \tag{A.2}$$

Note that $H$ and $\Sigma_\xi$ are PSD, as is $\Sigma_t$. Thus, the trace term is strictly non-negative for all time. □

**Lemma 5.** *Let $\rho(A) < 1$ with the total loss defined as previously (again with all eigenvalues of $H$ finite), and let $A$ be diagonalizable. Then,*

$$L(\theta; T) < \infty \tag{A.3}$$

*for all $T > 0$.*

*Proof.* First, note that $L(\theta; T) \leq L(\theta; T + 1)$ for all $T$ as both terms in the sum are non-negative. Thus, it suffices to show $\lim_{T \to \infty} L(\theta; T) < \infty$. We will address the two terms in the sum in (8) in order. First, note that

$$A^t = V \Lambda^t V^{-1} \tag{A.4}$$

where $V$ is a matrix with columns corresponding to eigenvectors of $A$ and $\Lambda$ is a diagonal matrix containing the eigenvalues of $A$. We will write the stacked eigenvalues of $A$ as $\lambda \in \mathbb{R}^N$. Then, we can rewrite the first loss term as

$$\phi_0^\top (A^t)^\top H A^t \phi_0 = z_0^\top (\Lambda^t)^\top V^\top H V \Lambda^t z_0 \tag{A.5}$$

$$= (\lambda^t)^\top Z^\top V^\top H V Z \lambda^t \tag{A.6}$$

where $z_0 = V^{-1} \phi_0$, $Z = \operatorname{diag}(z_0)$, and where the second line uses the fact that $\operatorname{diag}(z)\lambda = \operatorname{diag}(\lambda)z$. Thus, we have

$$(\lambda^t)^\top Z^\top V^\top H V Z \lambda^t \leq \|Z^\top V^\top H V Z\|_2 \sum_i |\lambda_i^t|^2. \tag{A.7}$$

Note that $|\lambda_i^t| = |\lambda_i|^t$. We will exchange the sums over $i$ and $t$ and note that, because $|\lambda_i| < 1$ for all $i$,

$$\sum_{t=0}^{\infty} |\lambda_i|^{2t} = \frac{1}{1 - |\lambda_i|^2} \tag{A.8}$$

for each $i$, and thus the sum over $i$ is finite. Thus, we have shown the first term in the loss is finite—not including the factor of $1/T$. As $T \to \infty$, considering the factor of $1/T$, this first term vanishes.

We now have to bound the second term in the sum over time in (8). Note that

$$\frac{1}{T} \sum_{t=1}^{T} \operatorname{tr}(H(\Sigma_t + \Sigma_\xi)) = \frac{1}{T} \sum_{t=1}^{T} \operatorname{tr}(H\Sigma_t) + \operatorname{tr}(H\Sigma_\xi) \tag{A.9}$$

and $0 \leq \Sigma_t$, as it is PSD and symmetric by construction and thus this term in the loss is non-negative.

We now need to construct an upper bound. Note that showing the boundedness of the two-norm of $\Sigma_t$ implies the trace is also bounded. Moreover, we have

$$\|\Sigma_t\|_2 = \|\sum_{k=0}^{t-1} A^{t-k-1}(I-A)\Sigma_\xi(I-A)^\top (A^{t-k-1})^\top\|_2 \tag{A.10}$$

$$\leq \sum_{k=0}^{t-1} \|A^{t-k-1}(I-A)\Sigma_\xi(I-A)^\top (A^{t-k-1})^\top\| \tag{A.11}$$

$$\leq \sum_{k=0}^{t-1} \|A^{t-k-1}\|_2^2 \|I-A\|_2^2 \|\Sigma_\xi\|_2 \tag{A.12}$$

$$= \|I-A\|_2^2 \|\Sigma_\xi\|_2 \sum_{k=0}^{t-1} \|A^{t-k-1}\|_2^2 \tag{A.13}$$

and thus it is sufficient to show the boundedness of the sum on the right hand side. Note that $\|A^{t-k-1}\| \geq 0$ for all $t, k$, and thus

$$\sum_{k=0}^{t-1} \|A^{t-k-1}\|_2^2 \leq \sum_{k=0}^{\infty} \|A^k\|_2^2 = \sum_{k=0}^{\infty} \rho(A^k)^2 \tag{A.14}$$

Because $\rho(A) < 1$, we have $\rho(A^k) < 1$; this yields boundedness of the infinite sum by following the same arguments as the first part of the proof. $\square$

We have thus shown that stability of the nominal model implies non-divergence of the stochastic model, and that the deterministic model provides a lower bound on the performance of the stochastic model. So, based on the above results, instability of the deterministic model (in which the total cost goes to infinity [83] in the infinite horizon) implies infinite cost for the stochastic model and stability of the deterministic model implies convergent (non-infinite) cost for the stochastic model. Thus, we discuss the behavior of the deterministic system as a proxy objective for the behavior of the stochastic system.

## A.2 Proofs from Section 4

*Proof of Theorem 1.* First, note that (9) corresponds to positive semi-definiteness of $\alpha I + P$. Due to this, as well as $H > 0$, we have $\lambda_i((\alpha I + P)H)$ real and non-negative for all $i$. Thus,

$$\rho(A) = \max_i |\lambda_i(A)| = \max_i |I - \lambda_i((\alpha I + P)H)| \leq 1 \tag{A.15}$$

if $\lambda_i((\alpha I + P)H) \leq 2$ for all $i$. Due to the positive (semi-)definiteness of $\alpha I + P$ and $H$, we have

$$\lambda_{\max}((\alpha I + P)H) \leq \lambda_{\max}(\alpha I + P)\lambda_{\max}(H) \tag{A.16}$$

and thus

$$\lambda_{\max}(\alpha I + P) \leq \frac{2}{\lambda_{\max}(H)} \tag{A.17}$$

ensures (10). Due to the positivity of $\alpha$, we have

$$\lambda_{\max}(\alpha I + P) = \alpha + \lambda_{\max}(P) \tag{A.18}$$

which gives our desired result. $\square$

*Proof of Lemma 2.* When we combine the Hessian preconditioner $H^{-\frac{1}{2}}$ together with our preconditioner $P$ and nominal step size $\alpha$, the resulting dynamics are

$$\phi_{t+1} = \phi_t - H^{-\frac{1}{2}}(\alpha I + P)\nabla_t. \tag{A.19}$$

which yields a nominal term $-\alpha H^{-\frac{1}{2}}\nabla_t$. The second term is

$$H^{-\frac{1}{2}}P\nabla_t = H^{-\frac{1}{2}}PH^{\frac{1}{2}}H^{-\frac{1}{2}}\nabla_t \tag{A.20}$$

where, note, $H^{-\frac{1}{2}}PH^{\frac{1}{2}}$ is a similarity transformation of $P$ due to the positive definiteness of $H$, and thus

$$\phi_{t+1} = \phi_t - (\alpha I + H^{-\frac{1}{2}}PH^{\frac{1}{2}})H^{-\frac{1}{2}}\nabla_t \tag{A.21}$$

with $\lambda_i(\alpha I + H^{-\frac{1}{2}}PH^{\frac{1}{2}}) = \alpha + \lambda_i(P)$. Plugging in for $\nabla_t$, we have $H^{-\frac{1}{2}}\nabla_t = H^{\frac{1}{2}}(\phi_t - \xi_t)$ which yields $\lambda_i(H^{\frac{1}{2}}) = \sqrt{\lambda_i(H)}$, given these results and following the bound on the maximum eigenvalue in Theorem 1 proves the result. $\qquad\square$

*Proof of Lemma 3.* Substituting for $P$ in Theorem 1, we have

$$-\alpha \leq \lambda_{\min}(\Delta\tilde{P}) \tag{A.22}$$

$$\lambda_{\max}(\Delta\tilde{P}) \leq \frac{2}{\lambda_{\max}(H)} - \alpha. \tag{A.23}$$

We have assumed $\Delta$ and $\tilde{P}$ are simultaneously diagonalizable, and thus

$$\text{diag}(\lambda(\Delta\tilde{P})) = V\Delta\tilde{P}V^{-1} \tag{A.24}$$

$$= V\Delta V^{-1}V\tilde{P}V^{-1} \tag{A.25}$$

for eigenvector matrices $V$, and thus $\lambda_i(\Delta\tilde{P}) = \lambda_j(\Delta)\,\lambda_k(\tilde{P})$ for all $i$ and some $j,k$. Note that because $\Delta$ is diagonal, the eigenvalues correspond to the diagonal entries.

We will consider the minimum eigenvalue bound first. Consider two cases. First, we will consider the $\lambda_{\min}(\tilde{P}) \geq 0$ case. As we know $\alpha > 0$ and $\lambda_j(\Delta) > 0$ for all $j$, this inequality is always satisfied. In the second case, $\lambda_{\min}(\tilde{P}) < 0$, the right hand side has

$$\lambda_{\min}(\Delta\tilde{P}) \geq \lambda_{\max}(\Delta)\lambda_{\min}(\tilde{P}) \tag{A.26}$$

$$= \max_i \overline{d}_i \lambda_{\min}(\tilde{P}) \tag{A.27}$$

Thus, to summarize, we have shown the above lower bounds the minimum eigenvalue of $\Delta\tilde{P}$ and thus satisfying it results in $\alpha + \lambda_i(\Delta\tilde{P}) \geq 0$ for all $i$.

We will now plug in to (10) with $P = \Delta\tilde{P}$. Note that, based on the simultaneous diagonalizability, $\lambda_i(\alpha I + \Delta\tilde{P}) = \alpha + \lambda_j(\Delta)\,\lambda_k(\tilde{P})$. Again, we consider two cases. If $\lambda_{\min}(\tilde{P}) \leq 0$, then the bound is immediately satisfied. Thus, the largest bound is provided by

$$\lambda_{\max}(\Delta\tilde{P}) \leq \lambda_{\max}(\Delta)\lambda_{\max}(\tilde{P}) \tag{A.28}$$

$$= \max_i \overline{d}_i \lambda_{\max}(\tilde{P}) \tag{A.29}$$

and plugging this into (10) yields the desired result. $\qquad\square$

## A.3   Robust Stability

Our robust stability analysis in Section 4.3 relies on a diagonal multiplicative error model, with close connections to D-stability [58]. There are many possible robustness conditions and forms of stability analysis within neural network models, most of which originate in the control theory and nonlinear dynamics communities [84, 85]. We highlight that a purely additive model (as opposed to the multiplicative model we use) would not necessarily result in the multiplicative factor in (16) and (17), and instead would result in an additive factor. Moreover, a term of the form $\epsilon\alpha H$ (as appears in our dynamics with cancellation) would not necessarily be present for an additive error model. Fundamentally, however, we believe our multiplicative error model is a reasonable one, with a combined multiplicative/additive model also being reasonable and general, and resulting in similar stability bounds.

## A.4   Non-Markovian Optimizers

Analysis of the stability of optimizers with momentum shows that the addition of momentum improves stability margins [56] and helps counteract noise [86]. We refer the reader to [56], Appendix B for an analysis of stability margins for Polyak and Nesterov momentum, and [87] for a comprehensive

exploration of the dynamics of Polyak-style momentum. This improvement in the stability margin is intuitive: if one considers the marginal stability case in which exact oscillation occurs on the sharpest eigenmode, momentum will act in the downhill direction at each timestep (for reasonably chosen hyperparameters) and move the system into a strictly stable regime. However, we emphasize that stability of the combined system requires analysis of the second order (double integrator) system induced by the interaction of the momentum hyperparameters with the stepsize parameters. This analysis is critical in the case of more expressive learned optimizers which may have recurrence. We believe this case—which is computationally and analytically challenging—is a primary direction of future work.

## B Experimental Details

In this section we provide details of the experiments featured in the body of the paper. Our experiments use JAX [88]. For all experiments we use PES [51] with a truncation length of 50, with a standard deviation of 0.01. We always target the mean training loss (across all timesteps of the inner optimization). In all experiments we use AdamW [89] as our meta-optimizer; this is identical to Adam other than how the weight decay is handled. We use zero weight decay except when explicitly stated otherwise, and apply gradient clipping of 1.0.

### B.1 Noisy Quadratic Model

In our NQM experiments, we considered a two-dimensional quadratic program in which

$$H = \begin{bmatrix} 1.11 & 0.596 \\ 0.596 & 0.486 \end{bmatrix}, \tag{A.30}$$

with the initial state $\phi_0 \sim \mathcal{N}(0, 10I)$ and the noise $\xi_t \sim \mathcal{N}(0, I)$ for all $t$. The curves in Figure 2a correspond to values

$$T \in \{1, 2, 3, 4, 5, 10, 25, 50, 100, 150, 200, 250, 300, 400, 500, 600, 700, 800, 900, 1000\}. \tag{A.31}$$

The plot visualizes the trace of the empirical gradient variance matrix

$$\hat{\Sigma} = \frac{1}{N} \sum_{i=1}^{N} (\nabla_i - \bar{\nabla})(\nabla_i - \bar{\nabla})^\top \tag{A.32}$$

where $\nabla_i$ is sample $i$ (with respect to the noise instantiation) of the gradient of the loss with respect to the parameters and

$$\bar{\nabla} = \frac{1}{N} \sum_{i=1}^{N} \nabla_i. \tag{A.33}$$

The figure is plotted for 50 values of $\alpha$ (with $P = 0$) at 500 seeds per sample.

In Figures 2b,c we consider a total episode length of $T = 50$. In this case, PES reduces to simple ES [8]. Plots show the mean over five seeds and the one standard error confidence intervals. In Figure 2b, the dotted line corresponds to the larger eigenvalues and the solid line corresponds to the smaller one. In Figures 2b,c we apply EMA smoothing with coefficient 0.95 to each curve (including the confidence intervals).

### B.2 In-Distribution Experiments

The two tasks considered for our in-distribution experiments are an MLP applied to Fashion MNIST and a ConvNet applied to CIFAR10.

The MLP model consists of two hidden layers with 128 hidden units each and ReLU activations. The ConvNet consists of three hidden layers with $3 \times 3$ kernels and ReLU activations. The layers of 32, 64, and 64 units respectively and strides of 2, 1, and 1 respectively. The conv layers are then followed by a fully connected layer. For both tasks, batch sizes of 128 are used. For more details on the tasks, we refer the reader to [9].

In our experiments, the "Hyperparam" optimizer corresponds to our architecture in which the blackbox output head is removed, and thus the optimizer corresponds to the nominal term with the

magnitude control head. The "Blackbox" optimizer corresponds to removing the nominal gradient term, which also corresponds to the *small_fc_lopt* optimizer of [9]. In our optimizers we use weight decay with AdamW, where the weight decay is specified as a multiplier on the outer learning rate. We use an outer learning rate of $10^{-4}$ for all of our experiments, for both tasks. The training horizon for the tasks are 2000 steps. In all of our experiments, we set $\beta_i = 10^{-3}$ for all $i$. In all plots, we apply EMA smoothing with a coefficient of $0.98$.

## B.3 Out-of-Distribution Experiments

We apply our optimizer, trained on the MLP/Fashion MNIST task, to a wide variety of other ML tasks. In particular, we apply it to 16 different tasks that span a wide range of architectures and datasets, from relatively similar to the training task—larger MLPs on CIFAR10, for example—to radically different tasks such as training models with ES gradients, recurrent models such as LSTMs, and transformers. These tasks are taken from the `learned_optimization` package [9], which is open source[6]. We add two baselines on these tasks: an Adam model and an NAdamW (Nesterov-accelerated AdamW) model. These baselines were also included in [9], and the tasks are all available in the learned optimization package. We note that the learning rate of the Adam model has been tuned for each task via a coarse search (half orders of magnitude steps), while the NAdamW model has been tuned for each task via random search. In particular, 1000 random hyperparameter values were evaluated for each task. Thus, the Adam baseline represents a reasonable baseline, comparable to standard training practice, while the NAdamW baseline represents an extremely strong baseline that is unachievable in typical hyperparameter tuning.

# C  Further Experimental Results

In this section we present further experimental results which extend and complement the results from the body of the paper. The details of these experiments are the same as previous unless otherwise stated.

## C.1  Alternative Views of Main Results

Figure App.1 shows individual seeds underlying Figure 3. There are two things to note about these figures. First, seeds which perform poorly often perform very poorly, diverging or oscillating around losses much worse than other seeds. This is to be expected from dynamical systems run for very long horizons, in which small amounts of instability are amplified over long timeframes. In spite of this, the results in Figure App.1 validate our results even if outliers are ignored.

## C.2  Ablations

Figures App.2, App.3, and App.4 show ablations of the various components of the STAR optimizer. Figure App.2 shows other values of weight decay in outer training. For values between 0.1 and 0.5 (times the learning rate) as we use in our experiments, a good tradeoff between stabilizing behavior outside of the training horizon and avoid over-damping the dynamics (as in the case of 1.0 weight decay) is achieved. We note that the poor training performance of 0.0 weight decay in Figure App.2b is due to an outlier that did not converge well in meta-training and low weight decay typically does not substantially harm meta-training.

Figure App.3 shows the impact of the Adam-style normalizing preconditioner. Generally, better meta-losses are achieved with the preconditioning compared to without; it is less clear what the benefit for generalization is, with performance of with/without the preconditioner being mixed. Generally, due to the strong improvements to in-task performance and limited degradation in out-of-distribution performance, we advocate for this term to be included.

Figure App.4 shows the impact of removing the nominal term. Generally, without the nominal term, meta-training is slower and converges to worse solutions. Interestingly, for the final trained optimizer, performance on the Fashion MNIST MLP seems comparable between all models; this is very much not the case for the CIFAR model. Generally, we have found performance at all points in

---

[6]https://github.com/google/learned_optimization

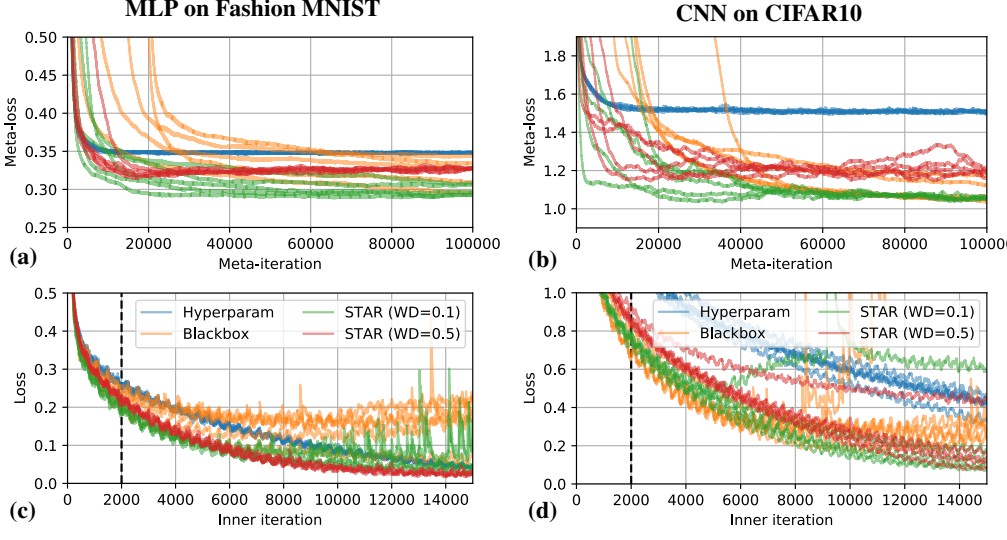

Figure App.1: Visualization of the individual seeds from Figure 3.

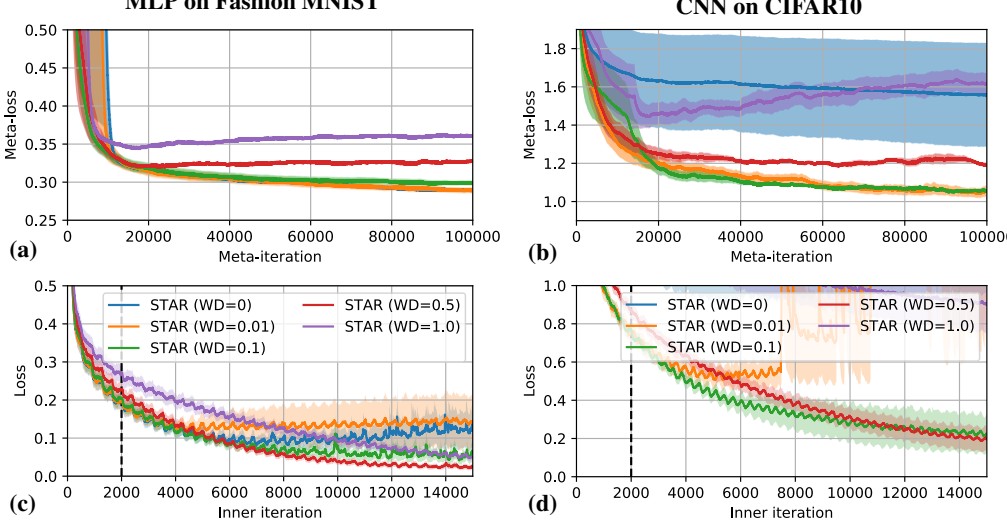

Figure App.2: Comparison of the impact of weight decay value for the STAR optimizer.

meta-training and generalization to be uniformly improved by the addition on nominal terms, and strongly endorse the inclusion of these terms.

### C.3 Full Generalization Experiments

We evaluate the learned optimizers trained on Fashion MNIST on a wide variety of tasks taken from the learned_optimization package, with results visualized in Figures App.5 and App.6. Note that both of these figures are the same other than the specific tasks evaluated, and they are split for space reasons. For a description of each task, see the source. Again, we note that the NAdamW baseline is an extremely strong baseline tuned to each task, whereas our model was trained only on the Fashion MNIST task.

We find in almost all cases that the blackbox optimizer is unstable as compared to Hyperparam, or either of the STAR optimizers. The performance of the hyperparameter controller versus the STAR models is more mixed; although we note that this degree of stabilization of blackbox models represents a significant advance and there are cases in which the STAR models substantially outperform the hyperparameter controller.

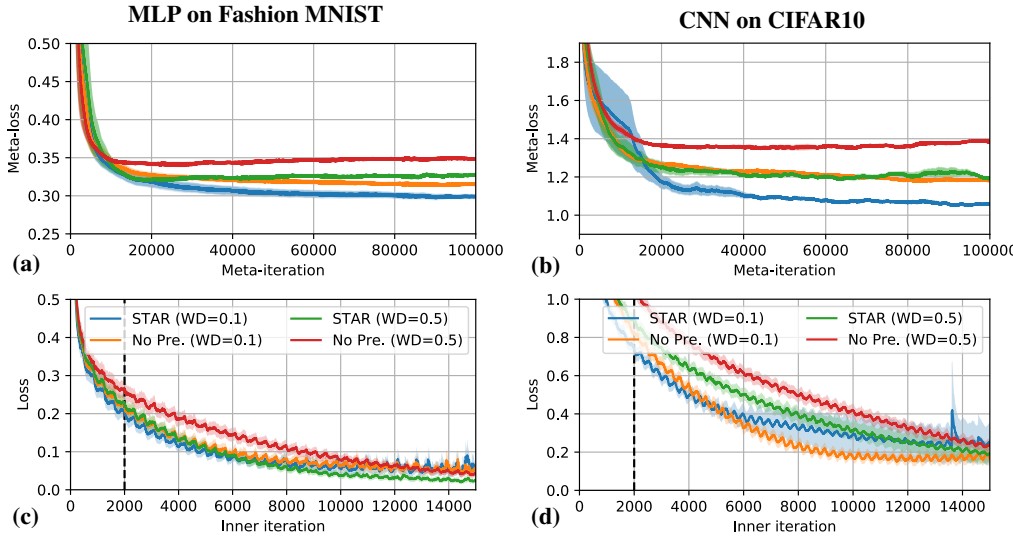

Figure App.3: Comparison of the impact of Adam-style preconditioning on the blackbox term. In these figures, "No Pre." denotes not applying the Adam-style normalization preconditioner.

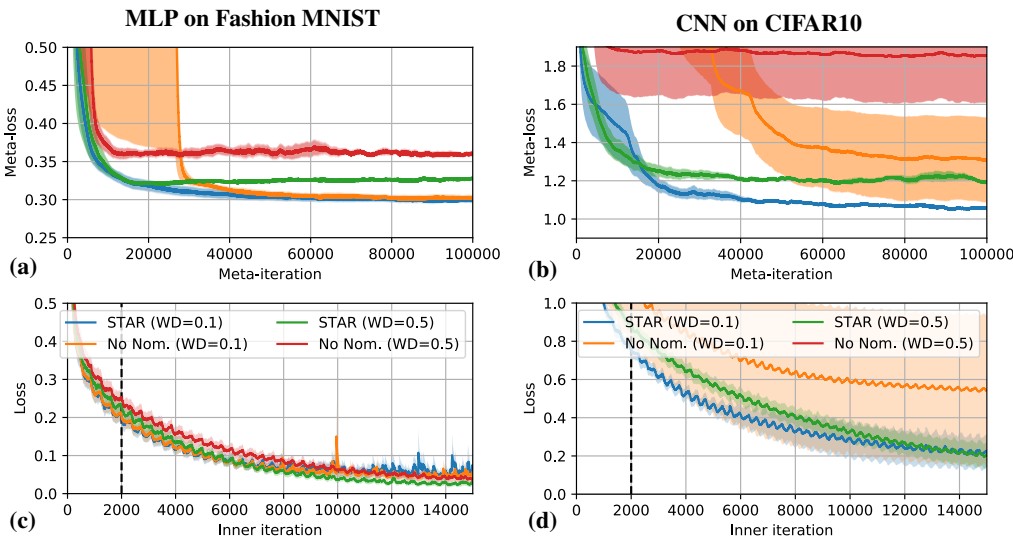

Figure App.4: Comparison of the impact of including the blackbox term. The "No Nom." curves correspond to only including a blackbox term.

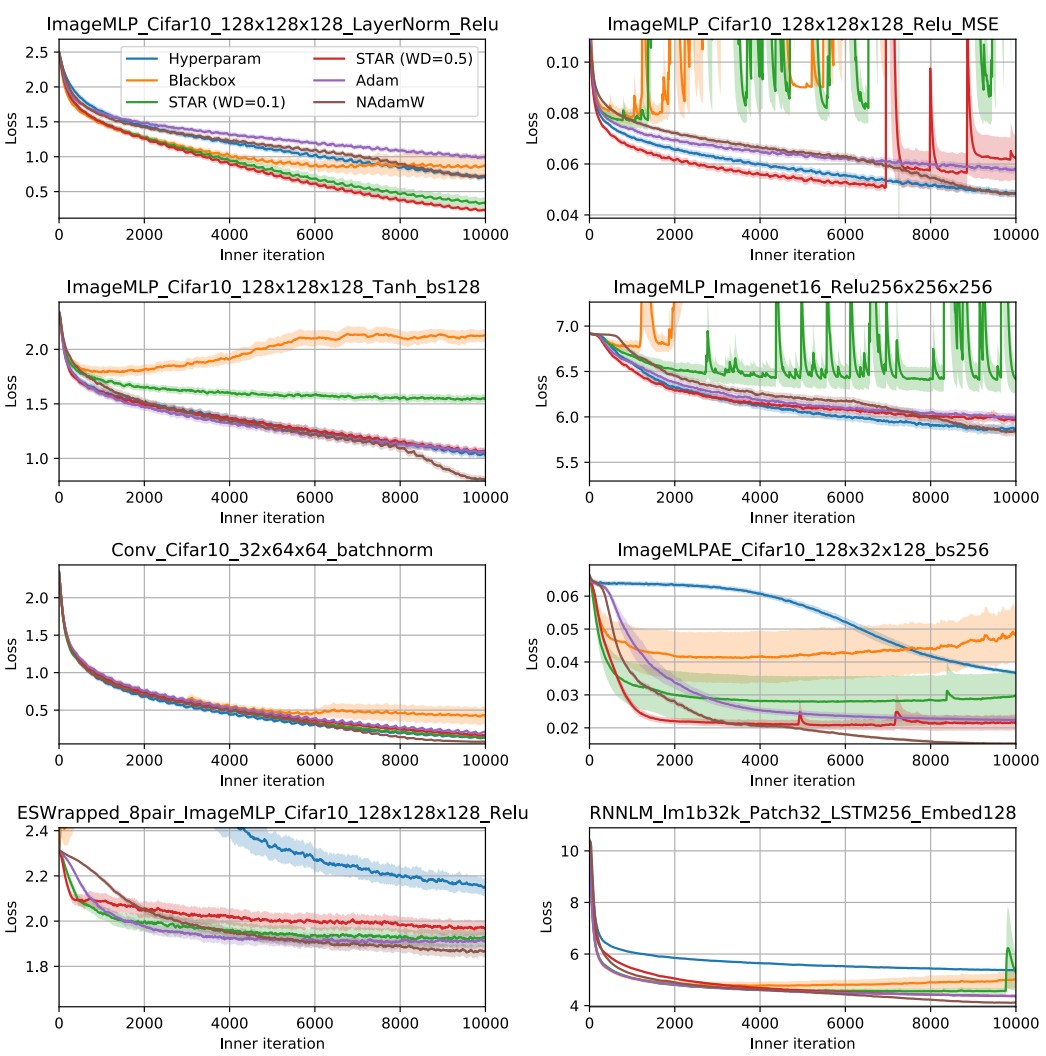

Figure App.5: Generalization results across a wide variety of tasks (indicated by subplot title).

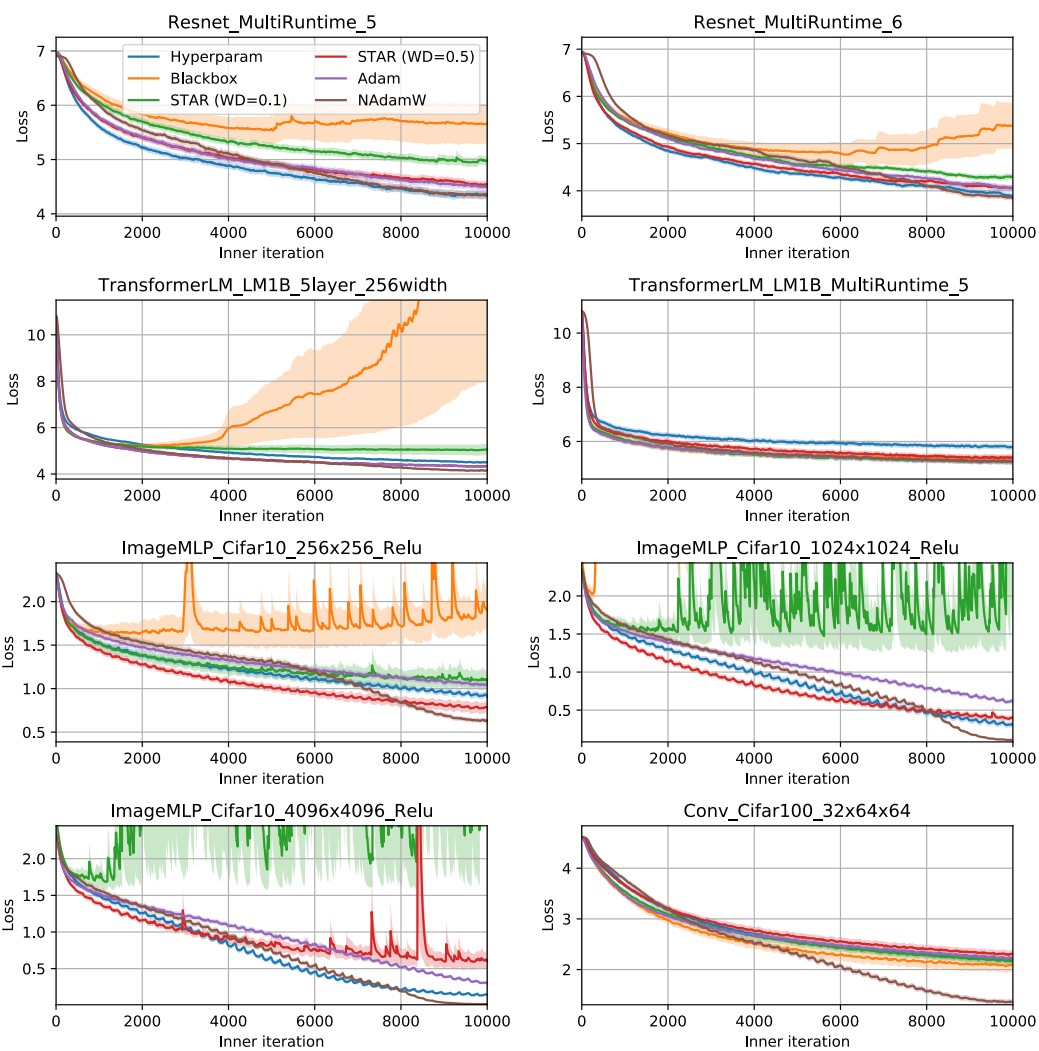

Figure App.6: Generalization results across a wide variety of tasks (indicated by subplot title).