# OpenReview forum: "A Closer Look at Learned Optimization: Stability, Robustness, and Inductive Biases"
_NeurIPS.cc/2022/Conference — NeurIPS 2022 Accept_

### Official Review · Reviewer_b5Ki · 2022-07-10

**Rating:** 7
**Confidence:** 3
**Soundness:** 3 good
**Presentation:** 3 good
**Contribution:** 3 good

**Summary:**

Learned optimizers suffer from being unable to extrapolate for a greater number of steps than they were trained on, and further the learning of these optimizers themselves are unstable. The authors propose that these learned optimizers can be studied under the lens of linear dynamical system, and by constraining the eigenvalues of the update map $A$ via introducing a nominal term, weight decay and preconditioning, the training of such learned optimizers become much more stable and are better able to generalize across greater time steps and even other domains.

**Questions:**

I was curious if learned optimizers did indeed diverge (or are more likely to diverge) when $\rho (A) > 1$. Do we find that this is true empirically for existing learned optimizers?

Generalization of the learned optimizer is interesting and I would expect the preconditioner matrix P and nominal term $\alpha$ provided by the learned optimizer to be domain dependent, and would be expecting more of the divergence in Fig. 4c, STAR (WD=0.1). Do you have an explanation why your method generalizes rather than diverge?

The Hyperparam experimental setting doesn't seem to be properly described in the paper, and it would be good if you could provide some details on that.

**Post Rebuttal** The authors clarified in detail the questions above, and I have changed my rating to reflect that.

**Strengths And Weaknesses:**

Strengths
- The perspective of viewing learned optimizers through the lens of linear dynamical systems is refreshing and enlightening, and could potentially explain why they diverge, and the authors provide theoretical substantiation as well
- Strong empirical results in both generalizing across time (in terms of additional time steps optimized) and across different domains

Weaknesses
- The flow of the paper is rough on the edges and could use a bit more intuition in some areas, particularly on the section on applying the preconditioning to the output vs to the input gradient
- Minor typos in the paper (e.g. "Incorporating *a* nominal terms" in pg. 3 Fig 1a, "to the *a* hyperparameter" in line 217, "inner *trainin*" in pg 8)

---

> ### Author Response · Authors · 2022-07-29
> **Reply**
>
> We thank the reviewer for their comments. We are currently addressing the writing issues raised by the reviewer, and will have a revised draft posted shortly. We will also discuss hyperparameter settings in more detail, as mentioned.
>
> The question of whether the theoretical conditions established in Section 4 are applicable to the models in Section 5 is a good one, and we also discuss this issue in our response to Reviewers nHvn and Bc3c. The conditions on rho(A) are sufficient to guarantee stability for linear systems. This form of stability analysis could allow us to make statements about the nonlinear system near an equilibrium point (i.e. around convergence). However, practically, we want to intervene to prevent divergence long before equilibrium is reached and thus statements about behavior near equilibria are of limited utility.
>
> Full stability analysis of the learned optimizer away from equilibrium will require nonlinear stability analysis. Tools for the stability analysis of large neural networks (or generic computational graphs) largely do not exist, and there are no tools that we know of that could be applied to our learned optimizer to design interventions. This is why we use motivation from the linear setting to guide the design of the STAR regularizations.
>
> We emphasize that the preconditioner used in the STAR optimizer (which is applied to the output of an MLP) is Adam-style normalization (diagonal preconditioner based on 1/sqrt of exponential moving average of the squared gradients). The strength of this normalization is indeed that it is domain dependent, and thus we automatically achieve rescaling based on e.g. layer width, which results in substantial stability improvements. The nominal term similarly has Adam-style normalization to improve generalization across domains, as well as meta-trained magnitude control for further generalization. If this did not sufficiently explain the observed generalization, we are happy to discuss further.

---

### Official Review · Reviewer_sccL · 2022-07-11

**Rating:** 6
**Confidence:** 4
**Soundness:** 4 excellent
**Presentation:** 2 fair
**Contribution:** 3 good

**Summary:**

This paper proposes a method to improve learned optimizer generalization by heavily regularizing both training and the architecture of the learned optimizer. These regularizations are derived from a theoretical analysis of the noisy quadratic problem from the perspective of dynamical systems. Using these insights, the paper presents experiments demonstrating the ability to generalize to much larger and very different problems, as well as much longer optimization depths.


**Questions:**

- What is the computational/memory overhead of the learned optimizer used? How does the training curve compare when plotted against training time instead of iteration/epoch?
- Is the blackbox magnitude term also an output head of the MLP, so the 3 heads are m_g, m_b, d?


**Limitations:**

- Evaluations are relatively limited in scale, with small architectures and datasets.
- The efficacy of this method in practice as an actual optimizer is questionable, though it seems positioned more as a path to eventual efficacy.


**Strengths And Weaknesses:**

Strengths
- The theoretical analysis provides good insight that leads to regularization techniques that will be useful for the area of learned optimization
- Experiments show reasonable improvements over a tuned Adam optimizer.

Weaknesses
- Evaluations are all on models much smaller than used in practice. The largest model used appears to be a “Resnet Like” ConvNet on CIFAR and a small transformer.
- The actual, exact regularizations that are constructed based on the theoretical analysis are far too obfuscated, given that this is the primary contribution of the paper. Tracking down a good summary of the actual method made takes quite a while and ends in the appendix, for example, section B.1 should probably be in the main text, since this is very important for readers who are considering applying this method.

---

> ### Author Response · Authors · 2022-07-29
> **Reply**
>
> We thank the reviewer for their comments. We will try to answer your questions and hopefully we can discuss further if necessary. We are working on a revised manuscript, which incorporates your feedback on paper structure (in particular, Section B.1). We will address two points in this response: the scale of evaluation tasks and the computational requirements.
>
> The reviewer has stated that the evaluations are much smaller than used in practice. We note that our largest evaluation task (the transformer model) has approximately 300 million parameters. This is almost 7x as many parameters as a ResNet101. While we do not test on extremely large models (such as billion+ parameter large language models), application on these tasks is a substantial engineering challenge requiring a shift to training over multiple accelerators. As the reviewer notes, we believe this paper is one step toward better learning optimizers, and toward an eventual goal of optimizers which strongly outperform standard optimizers on, for example, extremely large models like LLMs.
>
> The computational overhead is discussed briefly in Section 5 and Section B, and we will make this discussion more clear. The optimizer is nearly identical to that of [this work](https://arxiv.org/pdf/2203.11860.pdf). Indeed, as you note, we add one additional head to the output of the MLP for the magnitude control of the nominal term. This results in five additional weights, evaluated in parallel with the other MLP heads. The normalization of the blackbox output also has linear cost in the number of weights in the underlying model. Overall, compared to the expense of the rest of the model, the additional computational complexity over the baseline optimizer is negligible. We will add a more detailed discussion of additional induced complexity over the baseline to the paper. For plots of performance versus wall clock time, we refer the reviewer to the paper referred to previously, which contains experiments on this question (e.g. Fig. 1).

---

### Official Review · Reviewer_Bc3c · 2022-07-11

**Rating:** 7
**Confidence:** 4
**Soundness:** 4 excellent
**Presentation:** 4 excellent
**Contribution:** 3 good

**Summary:**

The paper studies the stability of the learning process from the lens of dynamical systems and characterizes the stability of the optimization process in terms of the eigenvalues of the training dynamics. They use the resulting insights to propose modifications to the learned optimizer’s architecture and the meta-learning procedure to improve its stability and generalization properties. They demonstrate the effectiveness of their modifications by testing the learnt optimizer on a variety of tasks. Importantly, they show that the resulting learned optimizer generalizes much better to tasks very different from the tasks it was meta-trained on.

**Questions:**

I think both the points in the Weaknesses sections are fixable and would appreciate the authors making amends along those directions.

**Minor Comments**
1. In line 299 of the paper, the authors say “STAR’s performance is comparable (and occasionally substantially better, 300 as in Figure 4a) to heavily hyperparameter-tuned Adam-based model”. From looking at the plots though, this sounds a bit disingenuous. STAR seems to be doing better on the CIFAR plots but worse on the other datasets (which is reasonable btw, I don’t expect it to work better, but I think it should be made clear in the text). So I would prefer if the paper talked about the differences in performance between tasks and the plausible causes.
2. Typos - line 217 “the a” -> “a”

**Limitations:**

The paper discusses certain limitations of the current approach w.r.t the assumptions regarding the convexity of the optimization problem and the need for baking in more inductive biases in order to generalize across tasks. However, I would like to see a deeper discussion on why the proposed learned optimizer still fails to outperform standard optimizers like Adam/NAdamW on a variety of tasks.

**Strengths And Weaknesses:**

**Strengths**
1. I liked the first principles approach of analyzing the stability of the existing learned optimizers presented by the paper. Moreover, the paper used the analysis to propose well motivated improvements to the architecture which resulted in significant performance gains.
2. The paper demonstrates impressive results on various tasks. The improvements on tasks that the optimizer wasn’t trained on were particularly impressive!
3. The paper was also clearly written and easy to follow.

**Weaknesses**
1. Ablation/analysis of the stability of the optimizer and validate if it was indeed the stability causing the issues.
2. The experiments show that the optimizer does reasonably well when tested on tasks very different from the tasks it was trained on. However, I would also like to see how well the optimizer performs when trained on a very wide range of tasks. Does it perform better than the standard optimizers (like adam) across all possible tasks? How much better/worse does it perform compared to an optimizer trained for the specific task? etc.

---

> ### Author Response · Authors · 2022-07-29
> **Reply**
>
> We thank the reviewer for their comments. In particular, we are glad that you found the out of (task) distribution generalization as remarkable as we did! We wanted to quickly address your points, and hopefully further discussion will be possible if time allows. We thank the reviewer for minor comments on the writing, and are preparing a revised draft. Our responses to the stated weaknesses are below.
>
> First, regarding linking the stability analysis to the divergence of the baseline blackbox optimizers. As discussed in our response to Reviewer nHvn, our linear stability analysis does not straightforwardly map to the nonlinear setting. While it is applicable around equilibrium points, the learning curves show that we are practically quite far from equilibrium over the course of training (losses are still going down). Thus, we do not expect to see the spectrum of the learned optimizers on the full problems to obey the conditions discussed in Section 4, and so we can’t explicitly identify exact violations of our theoretical results.
>
> In lieu of these experiments to link stability to divergence, we have included ablation experiments, which we believe partially connect the stability analysis to the observed performance improvements. For example, Fig. App. 2 shows a clear monotonic trend with respect to the weight decay, up until possible over-regularization for the WD=1.0 case on CIFAR10. The impact of the preconditioning on the output of the network (as visualized in Fig. App. 3) is limited on in-distribution tasks, but helps generalize across layer widths (as discussed in our response to Reviewer nHvn). Finally, the substantial impact of the nominal optimizer is clear from Fig. App. 4.
>
> While we agree that these results do not definitively connect our stability analysis to our interventions, exactly building this connection will require (potentially intractable) nonlinear stability analysis, which is beyond the scope of this paper.
>
> Second, on broader meta-training of the learned optimizer. We agree that such a training scheme is critical for meta-learning optimizers that can be deployed in practice. However, as discussed previously (in response to nHvn), the cost of this meta-training is extremely high. We agree that investigating this going forward is of high importance. We would note, however, that broad meta-training alone does not appear to be sufficient to achieve the strong out of distribution generalization that we achieve through our STAR regularizations (see e.g. the appendices of [this paper](https://arxiv.org/pdf/2009.11243.pdf) and [this paper](https://arxiv.org/pdf/1810.10180.pdf)).

---

> > ### Comment · Reviewer_Bc3c · 2022-08-08
> > **Response to author's comments**
> >
> > I appreciate the authors response. Thanks for the clarifications! I will keep the initial score!

---

### Official Review · Reviewer_nHvn · 2022-07-11

**Rating:** 5
**Confidence:** 4
**Soundness:** 3 good
**Presentation:** 3 good
**Contribution:** 2 fair

**Summary:**

This paper aims to find the inductive biases and properties for a good optimizer and hence improves upon the existing learnt optimizers.


**Questions:**

Can the authors present a reason why the optimizers fail to stabilize when the input is of high resolution as shown in Appendix

**Strengths And Weaknesses:**

Strenths
– An analysis of stability of the learnt optimizers.
– Designs new optimizer based on the properties that effect the stability.

Weakness
– Based on the definition of stability, convergence is not guaranteed. As the limit T->\inf L(\theta;T) can be finite even when the loss is oscillating.
– The results do improve on the existing optimizer, the results are shown on very limited setting.
– The results fail on high resolution imagenet and on bigger architectures.
– Due to the above reason, this is not useful in the practical settings.

---

> ### Author Response · Authors · 2022-07-29
> **Reply**
>
> We thank the reviewer for their review. We wanted to quickly try to clarify a few points.
>
> In the review, it is stated that the optimizers fail to stabilize on high resolution inputs. First, we note that while the stabilization may not be _complete_ (in that there is still some instability in training, or suboptimal convergence) our method needs to be understood as an intervention to the blackbox optimizer. Thus, in the generalization experiments in the appendix (Fig. App. 6), the STAR curves have to be compared to the blackbox optimizer without the STAR interventions (the orange curves). On the ImageMLP tasks (especially the larger ones), the orange curves nearly immediately diverge. Thus, in terms of the effectiveness of the intervention, we believe these results actually show the dramatic impact of STAR.
>
> We also wanted to explain why this behavior is happening. The sizes in the task titles in Fig. App. 6—for example, 1024x1024—actually refer to the MLP size (we are adding descriptions and code links to these models in the appendix). So, the 1024x1024 model has two hidden layers of width 1024. This width is the main cause of the instability. For very wide layers, small changes to the weights can result in very large changes to the pre-activations. Thus, it is important to decrease the step size used in optimization for larger models. However, our STAR optimizers (that we investigate generalization with) were only ever trained on hidden layer widths of 128. Thus, they never observe the need to decrease step size. The fact that our automatic step size tuning (based on Adam-style normalization on the output) even partially stabilizes the problem is thus a substantial improvement, and we expect this instability to be further mitigated by broader meta-training.
>
> We also wanted to try to better understand the stated weaknesses with regards to stability. The reviewer states that based on the definition of stability, convergence is not guaranteed. Are you referring to the linear/quadratic setting of Section 4? If so, this is a simple matter of exchanging our non-strict inequality (\rho(A) \leq 1) to a strict one (\rho(A) < 1). This should not meaningfully change any of the proofs in the paper. Indeed, while the current non-strict inequality can result in indefinite oscillation (marginal stability) as the reviewer notes, the set of optimizers with this property has measure zero.
>
> If you are referring to the definition of stability not guaranteeing convergence in the nonlinear setting, we acknowledge this shortcoming. However, full nonlinear stability analysis for neural network training is an extremely difficult question. While we believe that this is an important line of research, it is out of the scope of this first paper on the topic. Moreover, we note that much of the analysis of convergence for classical optimization algorithms is done using analytical tools that are very similar to our linear/quadratic analysis.
>
> Finally, we believe the experimental settings are not overly limited. The models we evaluate on span from two hidden layer MLPs to transformers with hundreds of millions of parameters. Indeed, Fig. 4d is the largest model we investigate (a transformer with approximation 300M parameters) and we find some of the strongest performance improvements of any problem for that model. While questions of larger-scale meta-training exist, this is hugely expensive and not easily included in this paper. Each meta-training run corresponds to effectively running a huge number of sequential training problems, and is thus not possible on very large models. Moreover, we emphasize that the goal of the paper is to propose the STAR interventions, which show broad performance increases on the wide variety of tasks considered.

---

### Author Response · Authors · 2022-07-29
**Comment for all reviewers**

We thank all of the reviewers for their thoughtful comments. We have tried to answer questions as promptly as possible, and we hope that our responses can be the basis for further discussion insofar as the time limits allow it. We are currently preparing a revised paper draft based on the comments on the writing (both structure and more minor changes).

---

> ### Author Response · Authors · 2022-07-31
> **Revised paper posted**
>
> We have now posted a revised version of the paper. We were not able to increase the discussion of the STAR optimizer in Section 5 in this revised draft due to space limitations. However, if the page limit is 10 pages for the camera-ready as in previous years, we will move content from Section B to Section 5 for the final paper. We have addressed all typos mentioned by reviewers and made a handful of other small writing changes. We softened our claims about the performance of the STAR model compared to baselines and improved the discussion around the Hyperparam model in the body of the paper (as requested by Reviewer Bc3c). We have also included a link to the public learned_optimization package where the tasks are implemented and documented. We will further discuss memory requirements (as requested by Reviewer sccL) and add details on the output normalization (as requested by Reviewer b5Ki) in the revisions to Section 5 for the camera-ready.

---

### Meta-Review · Area_Chair_EWif · 2022-08-25

**Recommendation:** Accept
**Confidence:** Certain

**Metareview:**

All 4 knowledgeable reviewers recommended acceptance of the paper (2x accept, 1x weak accept, 1x borderline accept), appreciating the the importance of the studied problem, the first principles approach and the obtained theoretical and empirical results. I mainly agree and recommend acceptance of the paper. Still, I ask the authors to carefully consider the reviewers' comments when preparing the final version of the paper and in particular improve the presentation in line with the suggestions. Also, some of the raised points on limitations should be included in a revised discussion.

**Award:**

No

---

### Decision · Program_Chairs · 2022-09-14

Accept